# *Trichoderma asperellum* Extract Isolated from Brazil Nuts (*Bertholletia excelsa* BONPL): In Vivo and In Silico Studies on Melanogenesis in Zebrafish

**DOI:** 10.3390/microorganisms11041089

**Published:** 2023-04-21

**Authors:** Adriana Maciel Ferreira, Iracirema da Silva Sena, Jhone Curti, Agerdânio Andrade de Souza, Paulo Cesar dos Santos Lima, Alex Bruno Lobato Rodrigues, Ryan da Silva Ramos, Wandson Braamcamp de Souza Pinheiro, Irlon Maciel Ferreira, José Carlos Tavares Carvalho

**Affiliations:** 1Research Laboratory of Drugs, Department of Biological and Health Sciences, Federal University of Amapá, Rod. JK, km 02, Macapá 68902-280, Brazil; 2Laboratory of Biocatalysis and Applied Organic Synthesis, Department of Exact Sciences, Chemistry Course, Federal University of Amapá, Rod. JK, km 02, Macapá 68902-280, Brazil; 3Central Extraction Laboratory, Graduate Program in Chemistry, Federal University of Pará, R. Augusto Corrêa, Guamá, 01, Belém 66075-110, Brazil

**Keywords:** Amazon fungi, *Danio rerio*, fungus extract, pigmentation

## Abstract

Endophytic fungi are those that present part of their life cycle in healthy tissues of different plant hosts in symbiosis without causing harm. At the same time, fungus-plant symbiosis makes it possible for microorganisms to synthesize their own bioactive secondary metabolites while in the stationary stage. To accomplish this, the endophytic fungus *Trichoderma asperellum* was isolated from *Bertholletia excelsa* (Brazil nut) almonds. The fungus was cultivated and extracted with ethyl acetate, obtaining AM07Ac. Then, using HPTLC (High-performance thin-layer chromatography) and nuclear magnetic resonance (^1^H NMR), β-amyrin, kaempferol, and brucine were identified as major compounds. Further in vivo assays in zebrafish demonstrated the activity of AM07Ac on melanogenesis by producing a concentration–response inhibitory effect, which, through an in silico study, proved to be related to the noted major compounds known to inhibit tyrosinase activity. The inhibition of tyrosinase prevents melanin accumulation in skin. Therefore, these results imply the importance of investigating microorganisms and their pharmacological activities, in particular the endophytic fungus *Trichoderma asperellum* as a generator of active metabolites for melanogenesis modulation.

## 1. Introduction

Endophytic fungi spend part of their life cycle within healthy tissues of diverse plant hosts in symbiosis without causing any harm [1,2,3]. At the same time, fungus-plant symbiosis makes it possible for microorganisms to synthesize their own bioactive secondary metabolites with low molecular mass during the stationary phase [4,5] and form in the presence of abundant precursors of primary metabolites, such as amino acids, acetate, pyruvate, and others [6].

The Amazon holds about 80% of the world’s biodiversity. In Brazil, several tree species native to the Amazon region, such as *Bertholletia excelsa* H. B. K., are intrinsic to the food culture of traditional peoples and the economic development of the country. Wood, fruits, oil, and almonds of these species are widely used as raw materials [7]. The oil extracted from almonds is rich in unsaturated fatty acids (oleic, linoleic, and linolenic) with nutritional relevance owing to the content of vitamins and minerals. These substances present in Brazil nut almonds have antioxidant activity, acting on free radicals and preventing premature cell aging, but they can also act against inflammatory diseases, cancer, arteriosclerosis, and others [8,9]. The Amazon rainforest with its hot and humid climate also favors the growth and symbiosis of fungi and other microorganisms [10].

In particular, these climatic conditions are favorable to the production of bioactive metabolites from endophytic fungi. Compared to higher organisms, these metabolites are renewable, and large-scale production of bioactive metabolites can be carried out using existing technology, such as changing the environment and optimizing cultivation conditions. From the perspective of environmental conservation, only a single, small removal of fungus from the natural environment is required [11,12].

Recently, enzyme inhibitors have been highlighted as substantial tools with pharmacological potential [13]. As an important example, different genera of fungi produce several bioactive compounds with anti-tyrosinase activity, including antibiotics, enzymes, enzyme inhibitors, and growth promoters. These can be applied in the agricultural, food, and pharmaceutical industries [14].

During melanogenesis, melanin is produced and stored in melanocytes that contain tyrosinase. Owing to the toxicity of intermediates in this pathway, such as 5,6-dihydroxyindole-2-carboxylic acid (DHICA) and quinones, this reaction is restricted to the interior of specialized organelles in melanocytes, the melanosomes [15]. Excessive melanogenesis can lead to skin darkening and abnormal hyperpigmentation, causing various dermatological problems, such as freckles, melasma, senile lentigines, and even skin cancer [16]. Melanosomes are lysosome-like organelles that contain all the enzymes necessary for the melanogenic process, such as tyrosinase (TYR), dopachrome tautomerase (DCT), and tyrosinase-related proteins 1 and 2 (TYRP-1 and 2) [17]. The maturation process of this organelle begins inside melanocytes and ends with the transfer of melanin granules to keratinocytes [18,19].

In recent decades, several phenolic compounds that inhibit tyrosinase have been studied [20]. These efforts have led to the discovery of inhibitors, such as arbutin, kojic acid, and hydroquinone [21,22]. However, the use of these agents has been limited owing to low stability, insufficient biological activity, side effects, high toxicity, and limited ability to penetrate tissue [22].

Studies demonstrate that fungi belonging to *Aspergillus*, *Paecilomyces*, *Agaricus*, and *Myrothecium* produce a range of compounds with inhibitory activity against tyrosinase. Another genus related to the production of compounds with anti-tyrosinase activity is *Trichoderma*, likely by combining enzymatic degradation of the cell wall with the production of different secondary metabolites [23,24]. Studies describe fungi of the genus *Trichoderma* spp., which produce secondary metabolites such as pyrones, terpenoids, steroids [10], gliotoxins, gliovirins [23,24], and some antibiotic peptides, known as “peptaibols” [25].

Among the secondary metabolites of fungi are compounds that can stimulate the production of tyrosinases, such as phenolic compounds, aromatic compounds, and metallic ions. Tyrosinase can oxidize a wide range of phenolic compounds, peptides, and proteins that comprise tyrosyl residues [25,26,27]. The main step in the oxidation reaction of phenolic compounds by tyrosinase can, however, be impeded by several inhibitory enzymes [28,29,30].

*Trichoderma asperellum* represents a potential source of underexplored bioactive substances. Therefore, both in vitro and in vivo tests are necessary to validate the biological activities of these substances. To perform such validation, a new model vertebrate has been introduced, zebrafish, popularly known as zebrafish (ZF). This is a freshwater fish belonging to the Cyprinidae family (teleost class), measuring between 3 and 4 cm and originating in India [31]. The ZF genome has been fully mapped, and it demonstrated 70% genetic homology with humans [32]. As such, it has become an experimental model for the study of genetic and biological mechanisms of numerous human diseases [33]. 

The high fertility rate and rapid development of ZF make it an ideal model to elucidate the molecular basis of several diseases [34,35], as well as screen for bioactive compounds, such as the presence or absence of melanin, the easily visualized pigment we intend to study in this work [36].

## 2. Materials and Methods

### 2.1. Collection of Plant Material and Isolation of Trichoderma asperellum Fungi

*Bertholletia excelsa* almonds (Figure 1) were collected in areas 1 (W 52°18′20.976″; S 0°33′44.44″) and 2 (W 51°57′53.338″; S 0°25′21.39″). The endophytic fungus used in this study was isolated from the almond and stored according to the protocol described by Holanda et al. [37]. 

Sample fragments were mounted on carbon tapes and visualized in a scanning electron microscope (SEM, model HITACHI—TM3030PLUS, Tokyo, Japan) at an accelerated voltage of 20 kV. Our access to SEM was provided by the Research Laboratory of Drugs of the Department of Biological and Health Sciences (DCBS) at the Federal University of Amapá (UNIFAP).

The endophytic fungus *Trichoderma asperellum* was identified conventionally and by molecular methods at the Pluridisciplinary Center for Chemical, Biological, and Agricultural Research (CPQBA) at the State University of Campinas (Unicamp), SP, Brazil.

### 2.2. Fungal Extract Preparation (AM07Ac)

The fungi were cultivated in a 500 mL Erlenmeyer flask containing 200 mL of malt medium (2%) at pH 7.0 under constant stirring in a rotary shaker (Solab, Piracicaba, SP, Brazil) for 8 days (28 °C ± 2 °C, 160 rpm). Inoculations were performed on four circular discs (0.5 cm in diameter) of solid culture medium to obtain the extracts. The growth of mycelial mass in a liquid medium was carried out in triplicate. After the cultivation period, fungal growth was stopped with the addition of ethyl acetate, followed by vacuum filtration and partitioning with ethyl acetate (3 × 50 mL). Excess solvent was removed by a rotary evaporator (Quimis, model Q344M2, São Paulo, SP, Brazil) at a temperature of 40 °C. The material was lyophilized (LS 3000, Terroni, São Carlos, SP, Brazil) to obtain the final mass of the dry extract of 0.45 g.

### 2.3. Fungal Species Identification

#### 2.3.1. Fungal Genomic DNA Extraction

Genomic DNA from the culture was purified using the phenol DNA extraction protocol described by Aamir et al. [38]. The amplification of TEF and Beta-tubulin marker genes was performed by PCR using the extracted genomic DNA as a template. The primers (synthetic oligonucleotides) used for PCR reaction were as follows: 728f/TEFlr for sample CPQBA 2615-22 DRM 02 complementary to the TEF and Bt2a region.

#### 2.3.2. Genetic Sequencing

The amplification product was column purified (GFX PCR DNA and Gel Band Purification Kit, GE Healthcare, Chicago, IL, USA) and submitted directly to sequencing using an ABI 3500XL Series automatic sequencer (Applied Biosystems, São Paulo, SP, Brazil). The primers used for sequencing were EF1/EF2, 728f/TEFlr, and Bt2a/Bt2b. Both genetic distance analysis and partial sequences of genes obtained from the above-noted primers were assembled into a consensus (single consensus sequence combining the different fragments obtained) and compared with the sequences of organisms represented in GenBank (http://www.ncbi.nlm.nih.gov) accessed on 18 August 2022 and CBS (http://www.westerdijkinstitute.nl/) databases accessed on 26 August 2022.

The sequences of microorganisms related to the unknown sample were then selected for the construction of the dendrogram. DNA sequences were aligned using the CLUSTAL X program [39] within BioEdit 7.2.6 [40], and genetic distance analyses were conducted using MEGA, version 6.0 [41]. The distance matrix was calculated using the Kimura model [42], and construction of the dendrogram from the genetic distances was carried out using the neighbor-joining method [43] with bootstrap values calculated from 1.000 resamplings using the software included in the MEGA 6.0 program.

### 2.4. Chemical Profile of T. asperellum Fungal Extract (AM07Ac)

#### 2.4.1. Characterization of Extracts (AM07Ac) by HPTLC and ^1^H NMR

The extract (AM07Ac) was analyzed by adapting the methodology described by Pinheiro [44], using the techniques of high-performance thin-layer chromatography (HPTLC) and proton nuclear magnetic resonance spectroscopy (^1^H NMR).

#### 2.4.2. Chemical Profiling by HPTLC

Chromatographic analysis was obtained in a robotized HPTLC system composed of application modules (Automatic TLC Sample 4—ATS4) and a photo documenter (TLC Visualizer—CAMAG (Muttenz, Switzerland)). WinCats 1.4.6 was used to process the chromatographic data.

#### 2.4.3. Sample Preparation and Application

The sample was prepared from the dilution of 10 mg of extract in 1.0 mL of methanol. Aluminum silica gel chromatoplates F-254 60 Å (Silicycle, Québec, QC, Canada) were used for the chromatographic analyses in the spray-band mode in aliquots of 50 µg/band (5 µL of solution) of extract solutions. In addition, 0.1 µg/band of the standards of (i) β-amyrin terpene (Sigma-Aldrich, Louis, MO, USA), kaempferol flavonoid (Sigma-Aldrich), and brucine alkaloid (Sigma-Aldrich) was inoculated. Kaempferol (Sigma-Aldrich) was also used to evaluate antioxidant potential.

#### 2.4.4. Chromatographic Procedures

Chromatoplates were eluted in a Camag glass tank (Muttenz, Switzerland) in an isocratic dichloromethane/methanol system (98:2) with a chromatographic path of 70 mm. Chromatoplates were derivatized with selective developer solutions for terpenes and steroids (vanillin-sulfuric acid 10%—VAS), alkaloids (tartaric acid and potassium iodide—Dragendorff), flavonoids (2-aminoethyl-diphenylborinate and polyethylene glycol 400—NP/PEG), and for antioxidant compounds (DPPH˙ 0.5%).

#### 2.4.5. ^1^H NMR Spectrum

Samples were prepared using 20 mg of extract solubilized in 600 µL of deuterated methanol (CD_3_OD) in a Bruker apparatus, model Ascend™ (Rheinstetten, Germany), operating at 400 MHz. TopSpin 3.6.0 software was used for data control and treatment, FIDs obtained were submitted to a Fourier transform with LB = 0.3 Hz, and pre-saturation sequences with low-power selective irradiation were used to suppress the residual signal of H_2_O. The spectra were treated manually, corrected at the baseline, and calibrated using the solvent’s residual signal as an internal reference, CH_3_OH—δ_H_ 3.30 [44].

The abundance of functional groups present in the classes of metabolites of interest was analyzed in the processing of FIDs, grouped, and normalized into specific regions (δ = 0.5–1.5/1.5–3.0/3.0–04.5/4.5–6.0/6.0–9.0/9.0–10.0).

### 2.5. Melanogenesis Studies in Zebrafish Embryos

#### 2.5.1. Experimental Animals

Zebrafish (ZF) of wild AB strain, both sexes, and aged approximately 8 months were purchased from Power Fish (Betta Psicultura—Itaguaí, Rio de Janeiro, Brazil). Specimens were conditioned in aquaria on the Zebrafish Platform of the Research Laboratory of Drugs at UNIFAP by undergoing an adaptation period of 40 days at a controlled temperature (23 ± 2 °C) and 12-h light/dark cycle. The project was approved by the Ethics Committee for Animal Use (CEUA) of UNIFAP under protocol number 006/2020.

#### 2.5.2. Protocol to Determine the Effect of Fungal Extract (AM07Ac) on Melanin Synthesis in Zebrafish

The reproduction test followed the recommendations of the Organization for Economic Cooperation and Development (OECD, 236)/(OECD, 2013). Eggs were collected from at least three groups to avoid genotypic variants, following the technique described by Yang et al. [45]. The eggs were washed with aquarium water, randomized, separated into 7 groups with N = 50 eggs, and treated in triplicate. The groups were treated with *T. asperellum* fungal extract (AM07Ac) at concentrations 4.8, 15, and 30 mg/L and transferred to wells of a 96-well plate containing a final volume of 250 µL. Solutions of PTU (N-phenylthiourea) 25 µM (Sigma-Aldrich, Louis, MO, USA) and kojic acid 25 µM (Sigma-Aldrich, Seoul, Korea) were used as positive controls, while aquarium water and DMSO at 3% were used as negative controls. Microplates with the treated and control groups were maintained at a controlled temperature of 28 ± 2 °C in an incubator (SOLAB, Piracicaba, SP, Brazil). Teratogenic alterations, vitality, and absence of pigments were observed through analysis in an optical microscope at 24, 36, 48, 60, 72, 84, and 96 h post-fertilization (hpf) (OECD, 2013).

#### 2.5.3. Toxicity of Melanogenic Inhibitors

ZF embryo growth patterns were monitored at intervals of 24, 48, 72, and 96 hpf to determine the potential toxicity of melanogenic inhibitors. In the test, 200 embryos were used for each treatment to evaluate embryonic mortality, morphological malformations, and heartbeat disturbances.

#### 2.5.4. Analysis of Pigmentation in Zebrafish

ImageJ (Fiji distribution, version 1.52p, National Institutes of Health, Bethesda, MD, USA) was used to analyze the degree of pigmentation in zebrafish embryos. The images were first converted to 8-bit gray images, and the threshold was set to select only the pigmented area. The particle parameter was employed with an appropriate pixel size threshold to remove artifacts, such as eyes and shadows around the yolk sac, from the images. For the Nile Red experiment, the integrated density of each image was measured as Nile Red fluorescence intensity using ImageJ, following the methodology used in a previous study [46].

The proportion of melanocytes was determined with ImageJ software (NIH), using equal-sized boxes for the dorsal view of whole embryos. Quantitative values were calculated as a percentage of black proportion per whole image. To assess the significance of differences between the control and experimental groups, all statistical data were obtained from one-way ANOVA with Dunnett’s post-test using IBM SPSS Statistics Data Editor software (Version 19). The significance level was set at * *p* ≤ 0.05 versus the DMSO control group, and the data were represented as the means ± SEM (standard error of the mean). To calculate the percentage of melanocytes in ZF embryos, photographs of the dorsal and integral view of the embryos were first taken, and then the images were used for the quantitative measurement of melanogenesis in each ZF embryo (Figure 2) [47].

### 2.6. Optimization and Molecular Docking to Identify Secondary Metabolites

The 3D crystal structure of human tyrosinase (PDB ID: 5M8N) was downloaded from the RCSB Protein Data Bank (https://www.rcsb.org/) accessed on 18 August 2022, with a resolution of 2.60 Å and used in the molecular docking study [48]. The secondary metabolites identified in the composition of the AM07Ac extract (β-amyrin, kaempferol, and brucine) were drawn, and their bioactive conformation energies were minimized in ChemSketch software by the molecular mechanics method (MM+) with a force field based on CHARMM parameterization [49,50,51]. The receptor protein was prepared using Discovery Studio software (ACCELRYS, 2008) to remove water molecules and heteroatoms. The positive control ligands used in the molecular docking simulations were mimosine (MMS), PTU, and kojic acid.

Pyrx (version 0.8.30) was used as a graphical interface to couple the receptor protein with the extracted secondary metabolite ligands and positive control groups. The grid center coordinate parameters were defined as −30.7782, −4.4638.034, and −23.0952, according to the positive control MMS. To obtain greater computational precision, exhaustiveness parameter 8 was generated for the receptor, and the conformation with the highest affinity was selected as the final pose to be visualized in the software in Pymol 2.5.0 [52].

### 2.7. Statistical Analysis

Results of the area under the curve (AUC) application were organized as mean ± SD (standard deviation) and presented in graphs. Analysis of variance (one-way ANOVA) was performed, followed by Tukey’s test for multiple comparisons. Results that showed differences of *p* < 0.05 were considered statistically significant between the treatment and control groups.

## 3. Results

### 3.1. Identification and Phylogeny

Fragments of TEF and beta-tubulin genes were successfully amplified and sequenced from the genomic DNA extracted from the sample. The genetic distance analysis (Figure 1) recovered sample CPQBA 2615-22 DRM 02 in a cluster with a resolution of 98% with the strain Type CBS 433.97 for the species *Trichoderma asperellum*.

Thus, the results of the analyses carried out in the databases and the phylogeny suggest the final identification of sample CPQBA 2615-22 DRM 02 as *Trichoderma asperellum* Samuels, Lieckf. & Nirenberg [53]. Partial sequence of the samples is shown in Figure 2.

### 3.2. Chemical Profile of the Fungal Extract of Trichoderma asperellum (AM07Ac) by HPTLC and ^1^H NMR

#### 3.2.1. Characterization by HPTLC

Analysis of the chemical profile by HPTLC allowed evaluation of the chemical complexity of the extract (Figure 3). When a 10% sulfuric acid (VAS) vanillin solution was used, the extract showed a reaction indicative of a diversified composition of terpenes and steroids (Figure 3a). By using the NP/PEG reagent and observing the chromatoplate under 366 nm radiation, formation of green bands typical of flavonoids could be visualized, as also seen with the standard used, kaempferol (Kae), in addition to the characteristic blue color of other phenolic compounds (Figure 3b). A thin orange band also indicated a positive reaction for alkaloids (Figure 3c).

The tests also showed the ability of AM07Ac extract to sequester DPPH radicals (purple) through the formation of yellow bands (DPPH in molecular form). Results indicate the antioxidant capacity of the constituents present in the AM07Ac extract compared to the positive control kaempferol (Kae), as seen in Figure 3d.

#### 3.2.2. Characterization by ^1^H NMR

The chemical composition of the AM07AC extract was also analyzed through the abundance of functional hydrogen groups and chemical shifts observed in the ^1^H NMR spectrum. The region of the spectrum (Figure 4), corresponding to olefinic hydrogens *δ*H 4.5–6.0 ppm, had the highest signal intensity with 15% of the area, while the region of *δ*H 4.5–6.0 ppm, corresponding to signals from hydrogens linked to oxygenated carbons, had a peak intensity equal to 2.06% of the area. The *δ*H 1.5–3.0 ppm range of hydrogens bonded to unsaturated carbons had an area equivalent to 0.35%. Signals related to methyl, methylene, and methine group hydrogens at *δ*H 0.5 to 1.5 ppm obtained an area percentage of 0.16%, as shown in Table 1 [44].

Substances that were assessed positively in the AM07Ac crude extract on HPTLC were identified in comparison with ^1^H NMR spectroscopic data compared to reports in the scientific literature. Correlation between HPTLC analysis and ^1^H NMR spectra resulted in identifying the chemical markers β-amyrin, kaempferol, and brucine.

The analysis indicated a double-doublet in the region of *δ*H 3325 ppm, characteristic of carbinolic hydrogen in 3β-OH triterpenes; a signal at *δ*H 4871 ppm, typical of olefinic hydrogen; and peaks at *δ*H 2202 ppm, indicating allelic hydrogen and *δ*H 8228 ppm and confirming the presence of hydroxyl on the C-3 carbon of the A ring, allowing us to suggest a triterpene structure with a bear skeleton equivalent to β-amyrin (Figure 5) [44].

Spectroscopic analysis also indicated a pair of doublets at *δ*H 6372 and 6410 ppm of meta-positioned hydrogens, characteristic of the AB system of flavonols, and two pairs of doublets, *δ*H 7193 and 7231 ppm, typical of a para-substituted aromatic ring. These chemical shifts allow for inferring the presence of kaempferol (Figure 5) in the crude extract [54].

The 1H NMR spectrum also indicates a singlet δH 8124 ppm, characteristic of hydrogen bound to the aromatic ring of indomethanolic alkaloids derived from the strychnine skeleton. Another crucial point is the presence of a singlet at *δ*H 3372 ppm, resulting in the deshielding of proton 12 by the ether ring of 7 members. Thus, it is possible to infer that it is the brucine alkaloid (Figure 5) [55].

### 3.3. Melanogenesis Studies in Zebrafish Embryos 

#### 3.3.1. Effect of Treatment with Fungal Extract (AM07Ac) on Melanin Synthesis in Zebrafish

Our ZF phenotype-based screening model demonstrated that developing melanophores inhibit melanin synthesis after treatment for 24 h (between 24 and 96 hpf). All extract concentrations significantly affected pigmentation in developing melanophores in ZF embryos (Figure 6) without developmental disturbances compared to control groups. The anti-melanogenic effect was compared using known melanogenic inhibitors, kojic acid and PTU [56,57], as positive controls. Treatment with AM07Ac extract in ZF embryos resulted in anti-melanogenic effects in a dose-dependent manner (Figure 6E–G). 

Treatment with the 30 mg/L concentration (Figure 6G) produced a remarkable inhibition of pigmentation in ZF embryos, similar to that of the PTU-treated group (Figure 6A) at 24 hpf.

In this study, melanogenesis in ZF embryos (Figure 7) was shown to be effectively inhibited. Treatment with AM07Ac extract produced a concentration–response effect with statistically significant differences when compared to control groups (*p* < 0.05). Effects of melanogenic inhibitors on tyrosinase activity and melanin synthesis in ZF embryos were observed, and both tyrosinase activity and total amount of inhibited melanin content after treatment with the fungal extract were evaluated to estimate inhibitory activities.

Results indicated a significant reduction in tyrosinase activity and total melanin content after treatment with 30 mg·L^−1^ of extract, demonstrating an inhibition potential similar to that of the melanogenic inhibitors PTU and kojic acid at 96 hpf. Kojic acid reduced the pigmentation of ZF embryos up to 48 hpf, while concentrations of 4.8 and 15 mg/L of the AM07Ac extract showed a lesser effect at 96 hpf. In particular, yolk sac pigmentation was inhibited after treatment with the extract at all concentrations.

#### 3.3.2. Toxicity of Melanogenic Inhibitors

No significant changes in mortality were observed in groups of ZF embryos treated with PTU 25 µM, kojic acid 25 µM, DMSO 3%, or AM07Ac at 4.8, 15, or 30 mg·L^−1^; however, the mortality rate at 30 mg·L^−1^ was higher compared to the other groups. Treatments did not show morphological malformations, except for the AM07Ac extract at 30 mg·L^−1^, which showed abnormality in the size of the yolk sac. In the heart rate test, kojic acid produced a slight disturbance.

### 3.4. Inhibition Potential of Secondary Metabolites of AM07Ac Extract on the Enzyme Tyrosinase

Molecular docking was used to evaluate tyrosinase inhibition through the secondary metabolites β-amyrin, kaempferol, and brucine, as identified in the AM07Ac extract. Protocol validation was conducted by superimposing the crystallographic structure of MMS (positive control) on the biological target in order to reproduce in the in silico model bioactive conformation similar to that of tyrosinase co-crystallized with MMS (PDB ID: 5M8N) (Figure 8).

Validation of the in silico model, as proposed in this study, was considered satisfactory owing to similarity between the crystallographic ligand pose and the molecular docking poses through the root-mean-square deviation (RMSD) value of 0.96 Å. According to the literature, the docking protocol is similar to the experimental model in RMSD values ≤2 Å [58,59,60,61].

The interactions observed at the MMS binding site with tyrosinase (Figure 9A) are around the α-helix (between amino acid residues His381 and Ser394) and in the β-sheet (among amino acid residues Tyr362, Arg374, and Thr391). In MMS, hydrogen bonds with residues Tyr362, Thr391, and Ser394 can be observed.

Among the secondary metabolites evaluated, kaempferol (Figure 9D) showed the best binding affinity value (−6.8 Kcal/mol), followed by brucine (−5.7 Kcal/mol) (Figure 9E). These secondary metabolites have enough potential binding affinity energy to act as tyrosinase enzyme inhibitors compared to positive controls (Figure 9A–C).

Kaempferol (C) and brucine (D) exhibited interactions similar to those observed in the positive controls MMS, kojic acid and PTU for the amino acid residues Arg374, His381, and Leu382. These interactions reproduce those described in the literature for anti-melanogenic activity in an in silico model [46,49,62], as shown in Table 2.

## 4. Discussion

Terpenes represent chemosystematic markers of the genus Trichoderma with chemical diversity of structures ranging from sesquiterpenes to diterpenes and triterpenes, such as the β-amyrin identified in the species *T. asperellum* [63]. 

The aglycone flavonoid kaempferol is a metal ion chelator with a complete conjugate linkage system, a tightly coordinated oxygen atom, and an acceptable steric configuration [64]. Kaempferol contains two sites that interact with metal ions: the 3-hydroxyl or 5-hydroxyl of the C ring and the 4-carbonyl of the C ring [65]. The compound has several pharmacological activities, including anticancer, antioxidant, and anti-inflammatory activities [66].

Alkaloid is another class of metabolite found in *T. asperellum* extract with the potential to influence the activity of the enzyme tyrosinase. Brucine has demonstrated anti-inflammatory and analgesic activity, as well as antitumor potential and antagonism toward arrhythmia with a negative inotropic effect owing to oxygen consumption in the cardiovascular system [67]. 

Tyrosinase is the rate-limiting enzyme in melanin biosynthesis. The process of melanin production starts with the hydroxylation of tyrosine to 3,4-dihydroxyphenylalanine (DOPA) and then the oxidation of DOPA to dopaquinone. Therefore, tyrosinase has been considered a critical target for the development of melanogenic inhibitors [49,68]. 

The inhibitory activity of secondary metabolites from species of the genus *Trichoderma* has been investigated, and the extracts of *Trichoderma atroviride*, *Trichoderma gamsii*, *Trichoderma guizhouense*, and *Trichoderma songyi* were all demonstrated as potential inhibitors of the enzyme tyrosinase [36]. The present study describes, for the first time, the in vivo anti-melanogenic activity of the crude ethyl acetate extract of *T. asperellum* (AM07Ac).

No significant difference was noted between PTU and kojic acid (*p* > 0.05). However, the results show a significant difference between the crude extract of *T. asperellum* (30 mg/L) and the positive controls PTU and kojic acid (*p* < 0.05). The results obtained in the melanogenesis assay in ZF embryos and molecular docking suggest that the inhibitory activity may be related to the chemical markers of the extract, kaempferol and brucine, which showed better binding affinity with the enzymatic site of tyrosinase.

Binding affinity results show selectivity for flavonols and indomethanolic alkaloids derived from the strychnine backbone. Reported studies show the anti-aging potential of a limited group of polyoxygenated xanthones, some of them with anti-tyrosinase activity. The inhibitory activity of enzymes related to skin photoaging (tyrosinase, collagenase, elastase, and hyaluronidase) was investigated for the first time for three simple hydroxylated xanthones considered tyrosinase inhibitors with IC_50s_ on the same order of magnitude, but lower than the IC_50_ obtained for the control kojic acid [68]. The enzymatic activity of kaempferol has been investigated in different methods. Kaempferol has shown an inhibitory and competitive effect on in vitro assays of the tyrosinase enzyme, with no statistically significant differences when compared to kojic acid [69,70]. In vivo analysis demonstrated that kaempferol promotes a cellular response induced in melanogenesis and melanocyte growth, regulating the quantity, maturation, and transport of melanosomes [71].

According to Lai et al. [49], L-tyrosine and candidate inhibitors of the tyrosinase enzyme do not interact directly with zinc ions; instead, their aromatic hydroxy and keto groups are linked by hydrogen bonding to the water molecule, which then binds to zinc ions. Additional interactions include tight aromatic stacking bonds with His381 and hydrogen bonds of its carboxylate group with Arg374 and Ser394. Potential inhibitors with better-defined electron density than L-tyrosine have better binding affinities with enzymatic sites, highlighting the importance of electron pair donor groups, such as hydroxyls and carbonyls of kaempferol.

In addition to its antioxidant activity and cytotoxicity, kaempferol’s tyrosinase inhibitory activity has been described against B16 melanoma cells [72]. Kaempferol inhibition acts through competitive binding with the enzymatic site of tyrosinase and the metabolite, preventing the catalytic oxidation of L-DOPA to L-DOPAquinone through catecholase activity. The free hydroxyl C-3 of the C ring of kaempferol has been indicated to have an important function in the molecular mechanism of tyrosinase inhibition, as evidenced through the molecular docking assay, which showed a hydrogen bond with the residual amino acid Tyr362 of tyrosinase (Figure 9D) [73,74].

## 5. Conclusions

In the present study, the chemical markers kaempferol, brucine, and β-amyrin present in the fungal extract of *Trichoderma asperellum* were identified. By using in vivo assays in zebrafish, it was possible to demonstrate the action of the extract on the melanogenic process, which is related to the major markers identified that act by inhibiting tyrosinase activity, as demonstrated in the in silico study. The results of the melanogenesis inhibition test in zebrafish embryos treated with the AM07Ac extract produced a concentration–response effect, demonstrating a specific relationship with the compounds identified in this extract, and the results imply the importance of investigating microorganisms and their pharmacological activities, demonstrating the potency of Amazonian diversity in terms of organisms that generate active metabolites, in addition to their preservation.

## Figures and Tables

**Figure 1 microorganisms-11-01089-f001:**
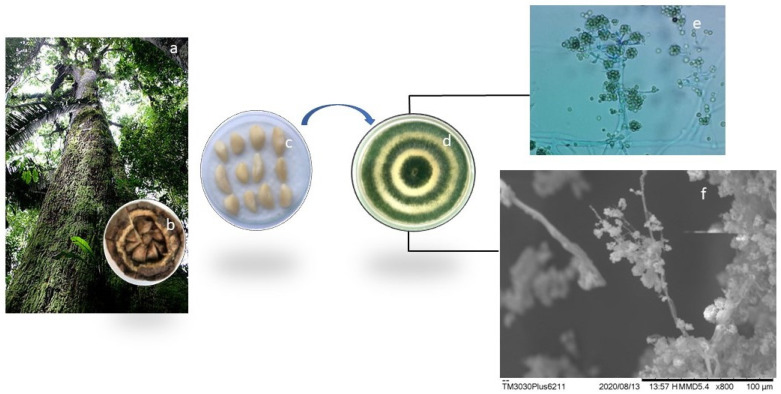
Tree of the species *Bertholletia excelsa* (**a**), hedgehog (**b**), and almonds (**c**), Morphological identification of the endophytic fungus *Trichoderma asperellum* cultivated in solid medium (**d**) with microculture observed by optical microscopy (**e**) and microstructural image by Microscopy Scanning Electronics (SEM) at 800× (**f**).

**Figure 2 microorganisms-11-01089-f002:**
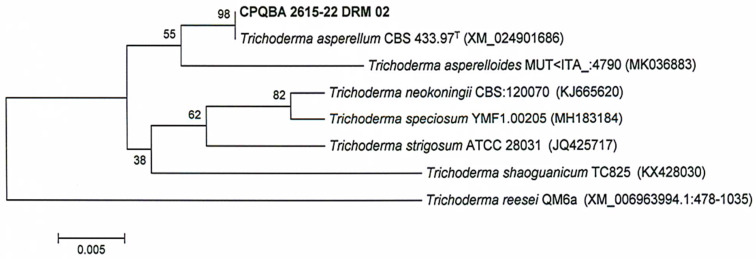
Dendrogram based on genetic distance using the neighbor-joining method to demonstrate the relationship between the partial sequence of the TEF region of the CPQBA 2615-22DRM sample and sequences of related microorganism strains present in the MycoBank (CBS KNAW, actual Westerdijk Fungal Biodiversity Institute) and GenBank databases.

**Figure 3 microorganisms-11-01089-f003:**
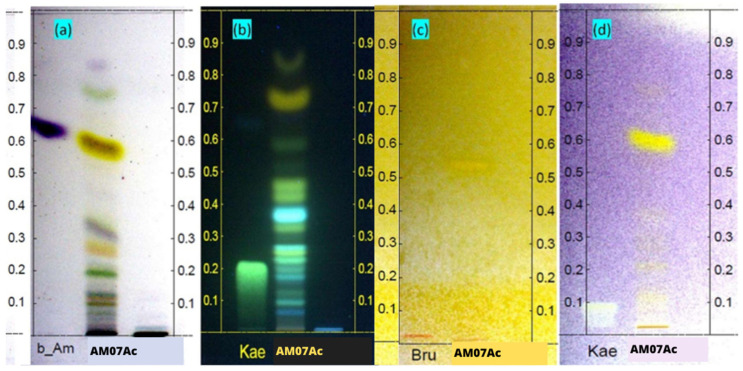
Chemical profile of extracts by HPTLC on chromatoplates derivatized with selective solutions for (**a**) terpenes and steroids in purple compared to the β-amyrin (b_Am) standard; (**b**) flavonoids in green and phenolic compounds in blue compared to the kaempferol (Kae) standard; (**c**) alkaloids in orange compared to the brucine (Bru) standard; (**d**) and antioxidant compounds in yellow compared to the kaempferol (Kae) standard. Standards were compared with *T. asperellum* crude extract (AM07AC).

**Figure 4 microorganisms-11-01089-f004:**
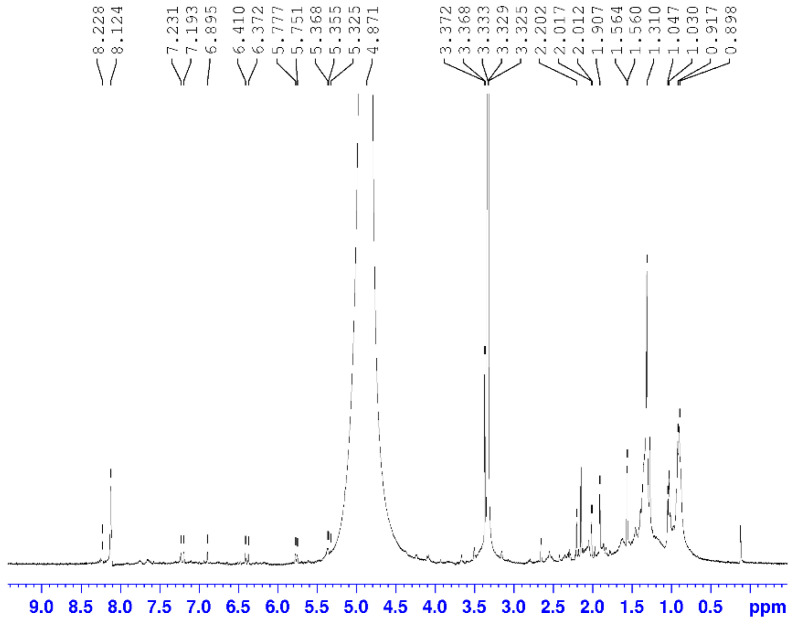
^1^H NMR spectrum of AM07Ac extract at 400 MHz.

**Figure 5 microorganisms-11-01089-f005:**
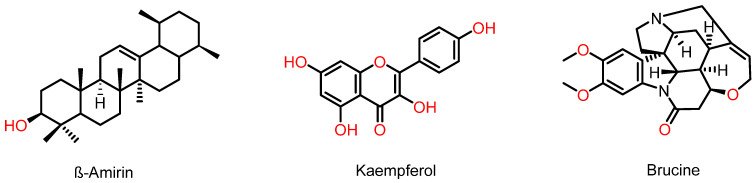
Molecular structures of secondary metabolites identified in the AM07Ac extract of the fungus *Trichoderma asperellum*.

**Figure 6 microorganisms-11-01089-f006:**
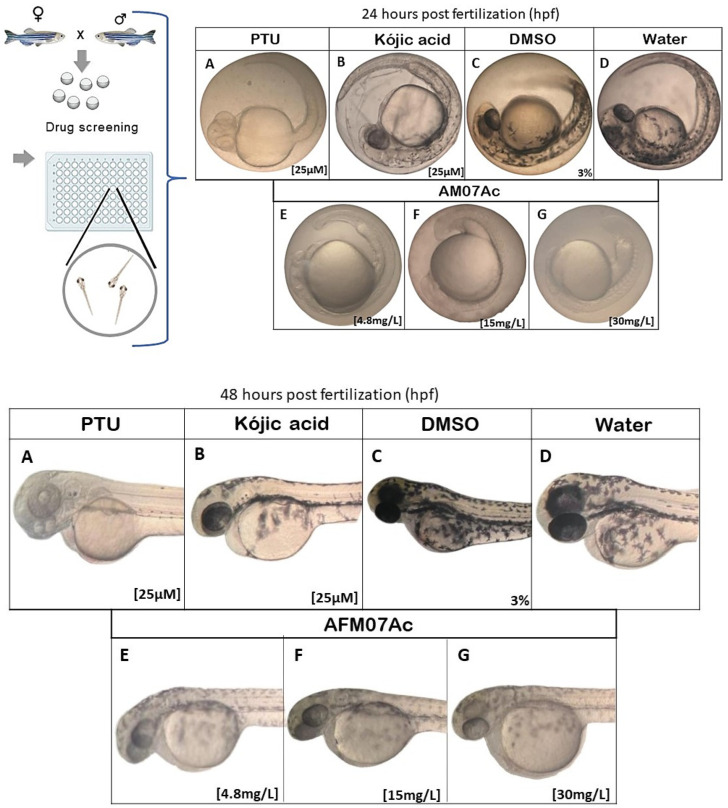
Effect of treatment with extract AM07Ac (**E**–**G**) on ZF embryos between 24 and 48 hpf. Embryos (**A**–**D**) were treated with positive and negative controls: (**A**) 0.25 µM PTU; (**B**) 0.25 µM kojic acid; (**C**) 3% DMSO; and (**D**) water.

**Figure 7 microorganisms-11-01089-f007:**
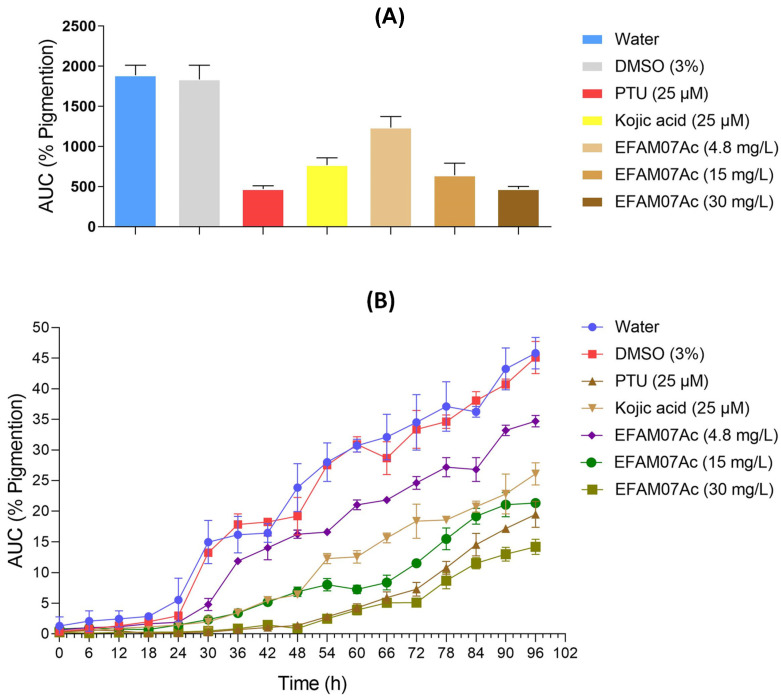
Effect of treatment with AM07Ac extract (4.8 mg·L^−1^, 15 mg·L^−1^, and 30 mg·L^−1^) on melanin synthesis and tyrosinase activity in zebrafish embryos. (**A**) Percentage of melanogenesis inhibition as a function of time in different fungal extract concentrations compared with positive and negative controls. (**B**) Expresses the area under the curve (%AUC) obtained by pigments in the embryos as a function of time. Results are presented as mean ± SD (n = 50 embryos/group). Differences between groups were analyzed by one-way analysis of variance (ANOVA), followed by Tukey’s test (*p* < 0.05).

**Figure 8 microorganisms-11-01089-f008:**
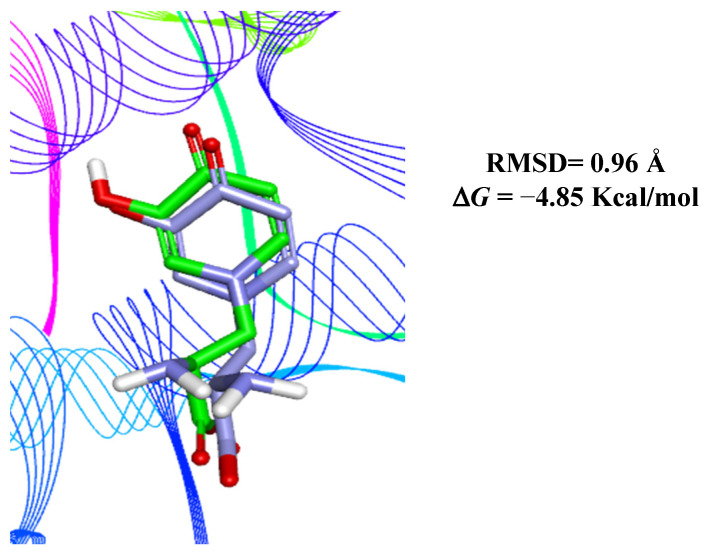
Superimposition of the docked MMS positive control (green) with the corresponding pose of the co-crystallized ligand in the enzymatic site of human tyrosinase (Lilac).

**Figure 9 microorganisms-11-01089-f009:**
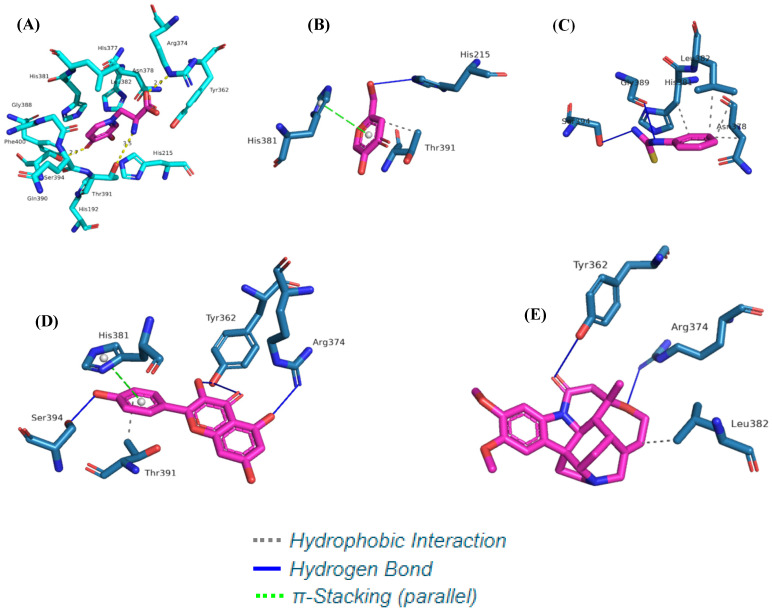
Hydrogen bonding or lipophilic interactions of positive controls MMS (**A**), kojic acid, (**B**) and PTU (**C**) and the secondary metabolites kaempferol (**D**) and brucine (**E**) identified in the extract (AM07AC) with the catalytic site of human tyrosinase.

**Table 1 microorganisms-11-01089-t001:** Spectroscopic characterization by ^1^H NMR of extract AM07AC.

Chemical Shift (ppm)	Assignments	Area (%)
0.5–1.5	–CH_n_; –CH_n_	0.16
1.5–3.0	CH_n_–C=O; CH_n_–N; Ar–CH_n_; Ar–CH_n_–	0.35
3.0–4.5	CH_n_–C=O; –CH_n_–O–; –CH_n_–N–	2.06
4.5–6.0	Ph–O–CH_n_; HC = C– (non–conjugated)	15.00
6.0–9.5	Ph–H; Ph–CH=O	0.13

**Table 2 microorganisms-11-01089-t002:** Results of binding affinities and their main interactions at the human tyrosinase receptor.

Protease	Binder	Binding Affinity (Kcal/mol)	H-Bond	Lipophilic Interactions
Tyrosinase	MMS	−5.9	Arg374, Thr391, Ser394	His192, His215, Leu382, His377, Tyr362, Asn378, His381, Gly388, Thr391, Asn378, Phe400
	PTU	−5.7	Gly388, Gly 389, Thr391	His381, Leu382
	Kojic acid	−5.7	His215	His381, Thr391
	Kaempferol	−6.8	Tyr362, Arg374, Ser394	His381
	Brucine	−5.7	Tyr362, Arg374	Leu382
	β-Amyrin	−4.5	-	Gln390

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
