# Peer review of "Trichoderma asperellum Extract Isolated from Brazil Nuts (Bertholletia excelsa BONPL): In Vivo and In Silico Studies on Melanogenesis in Zebrafish"

_microorganisms, 2023, doi:10.3390/microorganisms11041089_

Round 1

Reviewer 1 Report

The authors propose very interesting research, they studied an endophytic fungus with therapeutic potential.

I agree with the publication of the work and detail the aspects that were the basis of this decision:

- the work is written according to the magazine's requirements;

- the summary of the paper presents the purpose and objectives that were the basis of this study

- in the introduction, the authors include general data about the potentially beneficial role of endophytic fungi, they start from the idea that a fungus in a symbiosis with a species from the higher plant kingdom can help each other, it is possible that in the higher plant a quantity of higher than primary and/or secondary metabolites; also in this section, the authors also describe very briefly the biodiversity of species in the Amazon area that can be generators of new plant sources of importance for the medical field; the authors also present systematized data about the fungus Trichoderma asperellum; the objectives of the research proposed by the authors are to determine the effects of Trichoderma asperelleum extract isolated from Brazil nuts on melanogenesis in in vivo and in silico research;

- materials and methods are properly presented, in this section the authors describe in detail the protocol addressed in this research; the methods and reagents needed to obtain the extracts, the type of equipment used in the research are described; thus aims at obtaining the raw material, obtaining the extract, determining the genomic DNA of the isolated fungus, determining the chemical, pharmacological and pharmacotoxicological profile through non-invasive tests on Zebrafish, applied molecular docking studies for the identified secondary metabolites;

- the results are presented for each sub-stage of the conducted research; the phylogenetic data presented in summary and in the form of a dendrogram confirm the veracity of this type of mushroom; by HPTLC the authors highlight the presence of certain classes of secondary metabolites, including terpenes, polyphenols and alkaloid-type nitrogenous compounds; the authors also include the representative chromatogram; characterization by spectrometric methods, NMR confirmed the presence of compounds identified by HPTLC; evaluation of melanogenesis on Zebrafish embryos with the presentation of supporting photomicrographs; data on specific enzyme inhibitions induced by the extract; in silico studies are specifically on the molecular targets involved in melanogenesis;

- the discussions are related to the proposed objectives of the study and correlated with the data from the specialized literature;

- the conclusions are well-pointed and open opportunities for research on new species of fungi with therapeutic relevance;

- every bibliographic indication is in agreement with the research.

Author Response

The answers are attached.

Reviewer 2 Report

The paper entitled “Trichoderma asperellum extract isolated from Brazil nuts (Bertholletia excelsa. BONPL): in vivo and in silico studies onmelanogenesis in zebrafish” written by Adriana Maciel Ferreira and co-workers revealed the endophytic fungus Trichoderma asperellum as a generator of active metabolites for melanogenesis modulation. The languages in the whole manuscript are suggested to be greatly improved, meanwhile some descriptions are tediously long. Moreover, there are major points as shown below that could be greatly addressed to further improve the manuscript, which can be resubmitted in Microorganisms after major revision.

Author Response

The answers are attached.
